# Built environment and physical activity in adolescents: Use of the kernel density estimation and the walkability index

**Isabella Toledo Caetano**[1☯]*, **Rogério César Fermino**[2,3‡], **Renato de Oliveira Falcão**[4☯], **Paulo Roberto dos Santos Amorim**[1‡]

**1** Department of Physical Education, Federal University of Viçosa, Viçosa, Minas Gerais, Brazil, **2** Postgraduate Program of Physical Education, Federal University of Technology Paraná, Curitiba, Paraná, Brazil, **3** Postgraduate Program of Physical Education, Federal University of Paraná, Curitiba, Paraná, Brazil, **4** Department of Economy, Federal University of Viçosa, Viçosa, Minas Gerais, Brazil

☯ These authors contributed equally to this work.
‡ RCF and PRSA also contributed equally to this work.
* isabella.caetano@ufv.br

**Data Availability Statement:** The data underlying the results presented in the study are available at (https://data.mendeley.com/datasets/2w87n5fznt/

## Abstract

The availability of places for physical activity (PA) and the walkability of the neighborhood can impact the level of PA of adolescents. However, studies of this nature are still limited in Latin America. This study had two objectives: 1- using kernel density estimative, it investigated whether individuals living near PA places that are more intensely distributed than dispersed are more likely to be sufficiently active; 2—checked whether adolescents who live in neighborhoods with better walkability have a greater chance of being sufficiently active. Were evaluated 292 adolescents and PA was measured by accelerometry. Were measured five environmental variables for composing the walkability index. 98 PA points (places) were identified and destinations within these areas were geocoded and kernel density estimates (KDE) of places intensity were created using kernels (radius) of 400m (meters), 800, 1200 and 1600m. Using Logistic Regression, the association between the intensity of PA places (classified into quartiles Q1(smallest)—Q4(largest)) and the probability of being "Sufficient PA"; and the association between walkability (quartiles Q1(least)—Q4(highest)) and the probability of being "PA Sufficient " were estimated (p≤0.05). There were associations only for the intensities of places with the largest radius. Among adolescents who lived in places with higher intensity compared with lower intensity places: 1200m (Q3, OR 2.18 95% CI 1.12–4.22; Q4, OR 2.77 95% CI 1.41–5.43) and 1600m (Q3, OR 3.68 95%CI 1.86–7.30; Q4, OR 3.69 95%CI 1.86–7.30) were more likely to be "Sufficient PA". There were also associations for walkability, where those living in places with better walkability (Q4, OR 2.58 95% CI 1.33–5.02) had greater chances of being "Sufficient PA" compared to Q1. In conclusion, living in places with bigger densities and better walkability increases adolescent's chances of being "Sufficient PA".

draft?a=6a179060-d2be-458e-a61b-
e7e96791a9e3).

**Funding:** This study was financed in part by the
Coordenac̨ão de Aperfeiçoamento de Pessoal de
Nível Superior – Brazil (CAPES) – PNPD – CAPES.
The funders had no role in study design, data
collection and analysis, decision to publish, or
preparation of the manuscript.

**Competing interests:** The authors have declared
that no competing interests exist.

## Introduction

Globally the level of physical activity (PA) of adolescents is declining [1, 2] being this a great
concern to the public health [2, 3] once it increases the risk of chronic diseases in the adult-
hood [4, 5]. The construction of environments that support PA is a sustainable strategy to
encourage people to adopt or to increase the levels of PA [6].

Studies of the associations between PA and characteristics of the built environment in the
pediatric population have rapidly increased in the last two decades [7–9]. However, inconsis-
tent associations have been verified between the studies [7, 9, 10], mainly in countries with low
and middle income, where few studies are available [7]. A possible explanation, is that the
majority of the evidences in this area have been carried out especially in developed countries
of Europe, North America and Australia [7, 10, 11] and, may not be generalizable for countries
of low and middle income [7, 11].

In this context, walkability is an important characteristic of the urban environment and
measures how inviting an area is for pedestrians to access different parts of the city on foot or
by bicycle [12, 13]. Different walkability indexes were studied to verify the influence of the
built environment in the active behaviors of adolescents [14–17].

However, indexes developed specifically for adolescents were not found, most of the studies
adapted the index developed by Frank et al. [12] in the studies developed with the pediatric pop-
ulation. In Brazil, there is no evidence of the existence of an index that can be applied in the
whole national territory with unrestricted access data and low operational difficulty [18]. This
variety of indexes has been justified by the fact that both the attributes and their quantification
for creating the walkability index must be thought specifically for the target population [19].

The presence and the number of places intended for the practice of PA and leisure of the
pediatric population have been investigated between the studies [7, 20, 21]. However, there are
deficits in the literature regarding the association between destinations (places) and the PA of
individuals. These informations are limited to only classify the access to the places in a binary
form, indicating just the presence or the absence of the destination in a certain distance from
the house (*buffer*) [22, 23]. Binary measures do not take into account that a destination located
in the center of an area is not equivalent to one located on the edges of that area and analyzes
as if the effect were the same for both, which may disregard the more gradual change in places
that are accessible and those that are not [23–25]. Furthermore, these access measures do not
provide any indication of how these places present themselves in relation to each other, in
other words, if they are concentrated or dispersed [24].

The Kernel density estimative (KDE) is an spatial method that analyses the pattern of dis-
tributed points in the space [26]. Different from other approaches, this one has the capacity of
indicating how much the places of PA present themselves as more intensely distributed (con-
centrated) or dispersed based on the distance to a specific place [22, 24].

The Kernel density estimative involves placing a symmetric surface over each point, evalu-
ating the distance from the point to a reference place based on a mathematical function, and
summing the value of all surfaces for that reference location. This procedure is repeated for all
reference places [27]. The density estimative creates a statistical surface so that, for example,
there is an accessibility value measured by the density of the destination, mapped at each point
in the study area. It is typically considered a more refined spatial statistical model as it can pro-
vide an accessibility estimate for each point in the study area, and not just a binary response of
"presence" or "absence". Therefore, kernel density estimative constitutes a tool that can assist
researchers in analyzing the density (concentration) of AF places in a given region and thus
verify the possible influence on the PA of individuals. This information can be important for
urban planning in order to provide more structures for the practice and encouragement of PA.

Globally, the Kernel density estimative is not much explored in the evaluation of the environmental characteristics related to PA [25, 28], and the little evidence are from studies conducted in developed countries like Australia [25], Germany [28–30] and USA [23, 31] and, should not be extrapolated to countries with low and middle income. The density estimative has been used to examine environmental attributes, such as the access to recreational resources [32], stores and commercial establishments [25] in adults; and, urban measures such as public transport stations and open public spaces [29], recreational facilities, as green spaces, parks and playgrounds [28], exposure to green areas and behavior of PA [33] in children. The analysis was also applied to verify the access and availability of parks, recreational facilities and places for PA related to the socioeconomic status of adults [23, 31] and the relation between social deprivation and distribution of playgrounds for children and adolescents [30].

In Brazil, the density estimative has been applied in the health area, especially in nutrition studies, to verify the influence of the intensity of the food establishments distribution on the health of individuals [34–36]. Reviewing the literature, up until now it has not been found, any research that examines the relation between PA and Kernel density estimative of the places for PA in Brazil. Considering the shortage of data concerning this in the countries of Latin America, the presentation of a new approach to evaluate the built environment and that allows to establish relations with PA is important.

We believe that density estimative analyzes can help answer whether the intensity of destinations (PA places) near participants' homes is related to their PA level. Likewise, the walkability index can help to understand the relationships between adolescents' PA level and walking structures in the places where they live. We hypothesized that increasing levels of destination intensity (PA places), measured by KDE and neighborhoods with greater walkability, would be directly associated with the level of PA.

This study had two main objectives: 1- using kernel density estimative, it sought to investigate whether individuals living near PA places that are more intensely distributed than dispersed are more likely to be sufficiently active; 2—check whether teenagers who live in neighborhoods with better walkability have a greater chance of being sufficiently active.

## Methods

### Study design and participant

This cross-sectional study was carried out in 2019, with adolescents of both sexes, with age between 14 and 16 years, enrolled in the first year of high school, in public schools. For this analytical study, were evaluated the surrounding area of the adolescents residence.

For sample size estimation it was considered the established population of 968 (number of students enrolled in the first year of high school, in 7 public schools of the city), the estimated prevalence for PA practice at recommended levels of 50% [37], drawing effect of 1.1, interval 95% confidence. Based on these criteria, a minimum sample size of 305 adolescents was reached. The sample size, calculated a posteriori, allows detecting associations with an odds ratio greater than 2.3 with a power minimum of 87% for an alpha value of 5%. For this purpose, the G*Power software version 3.1.9.7 was used. The final sample consisted of 309 students, belonging to six public schools (five state schools and one federal). To obtain the representative sample, the students from each school were selected by drawing lots, based on the list of enrolled students [38].

### Declaration of ethics

The study was conducted according to the guidelines of the Declaration of Helsinki and approved by the Ethical and Research Committee involving human beings of the Federal University of Viçosa (CAAE 00925118.6.0000.5153).

The students had to present an Informed Consent Form and an Informed Assent Term duly signed by their legal guardians and by the adolescents to participate in the study.

## Characterization of the study area

Viçosa is considered a small town [39], located in the mesoregion of Zona da Mata, in the state of Minas Gerais, Brazil. Has an area of approximately 299.42 km$^2$ [40], with an estimated population in 2019 of 78.846 inhabitants [41].

According to information from the demographic census, Viçosa is composed of nighty nine census sectors in the urban region and eleven in the rural area [40]. The area of coverage of the study were the urban census sectors. Due to different methodological criteria for the data collection geo-referenced in the rural area, the census sectors of this region were excluded from the analysis.

In S1 Fig it is possible to visualize the study area.

## Independent variables–environmental measures

Two environmental measures: 1) related information on walkability; and 2) related information to the Kernel density estimate from the places for PA.

**Obtaining and preparing variables.** The information regarding the walkability and the places for PA were obtained from three principal sources: 1) Geo-referenced database from the census sectors for the municipality of Viçosa, related to the Census of 2010 [40]; 2) data from the software Open Street Maps (OSM) regarding the streets of the municipality (OpenStreetMap, 2020); and 3) data from the search system of Google Maps. Such data are publicly available. All the data were processed in the Software Qgis (version: Hannover– 3.16.10) and referenced to the datum Sirgas 2000, in the Plane Coordinate System, Universal Transverse Mercator System (UTM) Zone 23S.

*Geo-reference of the addresses evaluated.* For the evaluation of the surrounding of the residences it was necessary to perform the georeferencing of the addresses of the adolescents, included in the assessment form. With the aid of Google Maps, the address of each adolescent was located. The Google Street View (http://www.google.com/streetview) was used to confirm if the location corresponded to the number of the residence evaluated. With the coordinate values and the use of Google Earth it was possible to mark the points of each location in the satellite images and elaborate the archive Keyhole Markup Language (KML) with the points, that afterwards were inserted in the Qgis and converted in the format of shapefile.

## 1. Related information on walkability

In the present study we used the index of walkability proposed by Vegi et al. [18] and validated for Viçosa. The index used is composed by five environmental variables available for each census sector: density of intersection of streets, residential and commercial density, presence of sidewalks and public illumination, determined from the sum of the z-scores from each variable, by the Eq 1 [18]. The descriptive measurements of the variables that compose the walkability index are presented in the S1 Table.

$$[(residential\ density\ z-score) + (\%\ of\ presence\ of\ sidewalks\ z-score) \\ + (commercial\ density\ z-score) + (\%\ of\ presence\ of\ public\ illumination\ z-score) \\ + (density\ of\ intersections\ z-score)] \tag{1}$$

The mesh of streets of Viçosa was obtained from the service OSM (OpenStreetMap, 2020). Afterwards, were excluded streets and routes not much used by adolescents, as, rural roads

and service roads. After the exclusion, it was calculated the sum of the length in kilometers of all the streets in each census sector.

The connectivity of the streets was obtained from the road network, where feature analysis tools, available in Ggis, were used to identify and count the intersections of all streets. Subsequently, the number of intersections was divided by the total length of the streets from each sector, to obtain the density of intersections of the streets.

Using georeferenced data from the Census of 2010 [40] were calculated the residential and commercial densities. The number of residences from each census sector and the accounting of non-residential establishments (excluding garages, deposits, vacant lots, empty stores/ rooms, constructions, offices, factories, and agricultural commerce, civil engineering and autoparts commerces) were divided by the total length in kilometers of the streets from each one of the census sectors evaluated.

Finally, were used the datas from the surrounding area of the residences related to the presence of sidewalks and public illumination. These information were obtained by means of direct observations realized by IBGE technicians [40]. The Census researcher, still in the collection phase, verified if the face of the residence had or not path with pavement, destined for pedestrian circulation. In this way, the presence of pavements was determined by the division of the quantity of residences that had them by the total of residences of each census sector. In the same way, the researcher verified if in the other side existed at least one public lamp post. For the two observed characteristics, determined the percentual of presence in the sector dividing the total of residences that had the characteristic by the total of residences in the sector and then multiplied by 100.

After the capture of the information of the five environmental variables (density of intersection of streets, residential and commercial density, presence of sidewalks and public illumination) that compose the index, was calculated the walkability for each census sector, that was posteriorly attributed to each participant. Based on the values of walkability, the adolescents were divided in quartiles (Walkability Quartile 1, Walkability Quartile 2, Walkability Quartile 3, Walkability Quartile 4), in a way that the individuals in the inferior quartiles resided in census sectors with lower values of walkability and those located in superior quartiles lived in census sectors with higher values of the index.

## 2. Places for PA and related information on the kernel density estimate from the locations for PA

**2.1 Places for PA used to estimate kernel density estimate.** Were identified 98 places, public and private, possible for PA of adolescents: 12 public schools, 1 public university, 15 soccer fields, 1 sports complex, 12 sports courts, 3 gymnasiums, 4 social clubs, 10 gyms, 3 outdoor gyms, 2 green areas, 1 running track, and 24 plazas/playgrounds.

The Google Street View was used to certify the presence of the place of PA and Google Maps was used to extract the geographic coordinates from each location. Google Earth was used to identify the points of each location in the satellite images and elaborate the archive KML for the insertion in the Ggis and converted in the format shapefile.

**2.2 Calculation of kernel density estimate.** The Kernel density estimate is a technique of data smoothing, registered as point and geographically referenced, that transforms a sample of observation in a continuous surface, indicating the intensity of the individual observation in an area of influence [42]. The Kernel considers the events within its area of influence according with the distance of the point where the intensity is being estimated into its center, where it is localized the observation of interest [22]. In the present study, the observations are the points of PA.

The function of Kernel density estimate is given by the Eq 2 [43].

$$\hat{\lambda}_{(s)} = \sum_{i=1}^{n} \frac{1}{\tau^2} k\left(\frac{(s - s_i)}{\tau}\right)$$

(2)

Where: $\hat{\lambda}_{(s)}$ is the estimation of the Kernel density of a point in the place $s$; $s_i$ is the i-th point observed, k represents the kernel weighting function and τ is the bandwidth (radius).

Therefore, the Kernel density estimate in certain s point, will correspond to the sum of the estimated density for each observation, in case s is contained in the areas of influence of these observations [22, 43, 44].

In Fig 1 it is possible to visualize the process of density estimate in a surface, with overlap of various radius of influence. Through a layer regarding the map of the study area, continuous and outlined by a pixel-grid, the density estimates fits in a series of centralized cones in each characteristic of interest [25]. Each cell in the surface of the map is attributed a density estimate that is evaluated according to the proximity to the Center of the cone/kernel, in this way the further from the center of the cone, lower is the intensity of the density estimate [43–45]. In

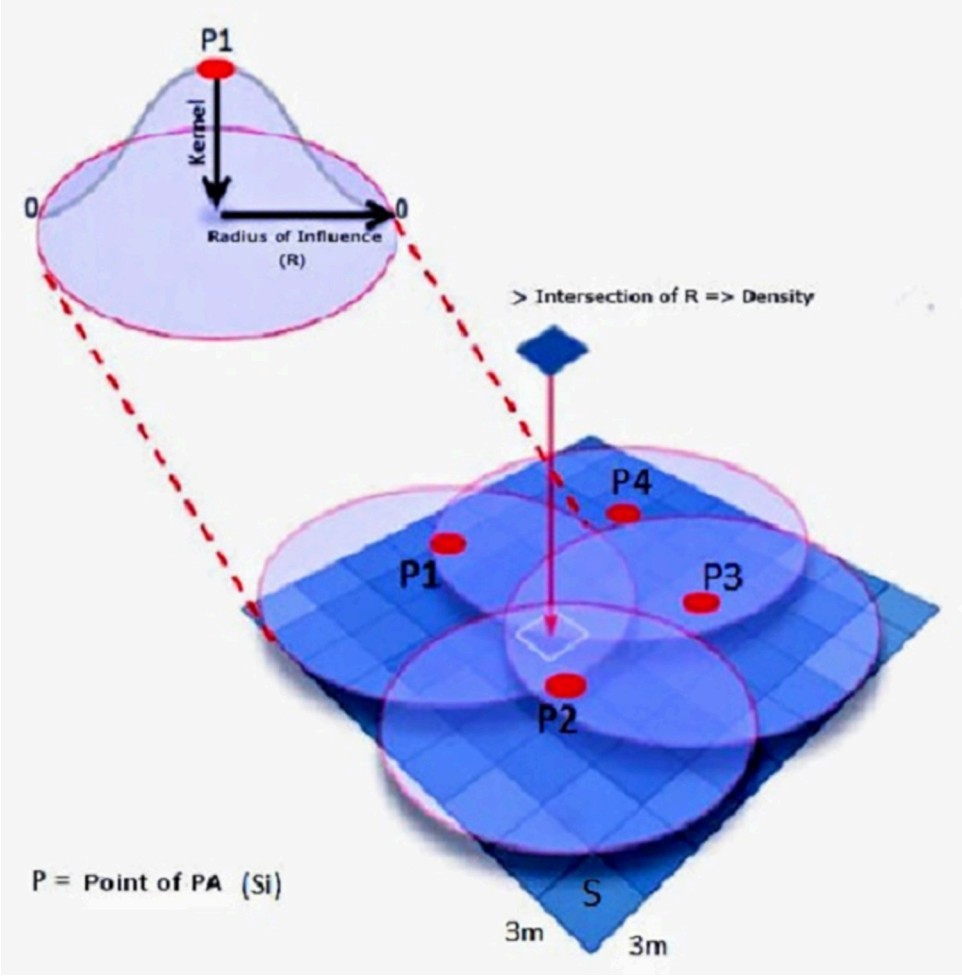

**Fig 1. Kernel density estimate.** Source: Adapted BERGAMASCHI [46]. Note: P, point of PA; R, radius of influence; m, meters; Si, i-th point observed; >, bigger.

effect, interest points located outside the radius of influence received density values equal to zero [43]. In addition, cones of different points can overlap, so the intensity of density in a determined region is given by the sum of Kernel density estimate of different cones overlapped [25, 44].

From the parameters used in the Eq 2, the quality of the estimative is defined by the choice of the kernel function and by the size of the radius [47]. The choice of the radius is more important for the quality of the Kernel density estimate [22, 43, 47].

Different sizes of radius were used in the studies with Kernel density estimate [24, 25, 28, 32]. In the present study, were determined four sizes of radius (400, 800, 1200 and 1600m (meters), resulting in four density surfaces (KDE): KDE for 400m radius, KDE for 800m radius, KDE for 1200m radius and KDE for 1600m radius.

In the Ggis, the estimate of the density was conducted using the tool "Heat Map". The estimative process in the software starts with the definition of the parameters of the Eq 2: layer of points that represent the places for PA ($s_i$, the size of the pixel, used to define each point where the density will be calculated ($s$, the kernel function ($k$) and the radius ($\tau$).

It is important to highlight that for density analysis it is important to determine a radius of influence around the point of PA. This radius can be assigned different sizes (for example, 400m, 800m), where it is observed that larger radius sizes produce more continuous density surfaces and at the same time encompass a greater number of residences close to the point of PA, making it possible that place of PA is available to a greater number of people.

In a hypothetical situation, a radius is drawn around each point of PA (place). Within this radius there are several density values calculated, so that the density values that are closest to the point of PA have greater intensity values and, as they move away from this point, their intensity is lower. In the case of two or more nearby points of PA, at the intersection of the areas formed by these rays, the density values are added, so that the more concentrated these points are, the greater the density intensities relating to those points of PA will be. Thus, if an individual's residence is located within one or more radius drawn based on the points of PA, it assumes the same density value determined for that location. From the density analysis of these places of PA close to the individual's residence, it is possible to verify whether or not there is an influence on their PA level.

The points of PA were inserted and combined in a single layer shapefile, so that the estimative was made according to the location of the 98 points. In this analysis were assigned pixels with dimensions of 3x3 meters and were used in the analysis of the square function [22, 43]. The result of the Kernel density estimate is a "Map of Kernel", in raster format, containing the values of the estimated densities. The descriptive measurements of the Kernel density estimate in the radius of 400, 800, 1200 e 1600m are presented in the S2 Table.

After the estimation of the Kernel density for four different superficies (KDE for 400m radius, KDE for 800m radius, KDE for 1200m radius and KDE for 1600m radius) was assigned a value for each individual. After, they were divided in quartiles (Q1 –Quartile 1, Q2 –Quartile 2, Q3 –Quartile 3 and Q4 –Quartile 4), in a way that those situated in the inferior quartiles had the lower values of density and those situated in the superior quartiles have higher density values.

## Dependent variable–evaluation of the physical activity

The accelerometer ActiGraph (model GT3X) was used to monitor the time spent in moderate to vigorous PA (MVPA) (min.day-1). The software ActiLife (version 6.13.4) (ActiGraph, LLC, Fort Walton Beach, USA) was used to perform all the analysis of the accelerometer. The adolescents used the monitors in the right hip in an elastic belt for 8 consecutive days, including

during the night sleep. The adolescents were instructed not to change their daily routine and the accelerometer should be removed only for aquatic activities. The first day of use (day that they received the equipment) was not considered in the analysis to avoid the Hawthorne Effect [48].

The accelerometer initialized to collect data with a sampling rate of 30 Hz, with normal filter, in epochs of 1s and then the data was reintegrated in epochs of 15s. The period of non-use was defined as zero counts/minute consecutive that lasted at least 20 minutes. To be included in the analysis, was necessary that the participants achieved a minimum of 10 h.day-1 of "time of use" [49], at least 5 days a week, in which, at least 1 day should be a weekend day. The analysis of time of sleep/wakefulness were done and removed from the analysis. To classify the PA were adopted the cut-off points developed by Romanzini et al. [50], validated for Brazilian adolescents, using magnitude vector and epochs of 15s. Based on the weekly average, the adolescents were classified in "Sufficient PA" when performed 60 minutes or more per day of MVPA or in "Insufficient PA" when they were below that time [37].

## Confounding variable

The socioeconomic status (SS) of the family was extracted from the Economic Classification Criteria of the Brazilian Association of Research Companies [51]. According to the final score, the participants' SS were classified into 3 classes: 'high' (classes A and B1), 'average' (B2 and C1) and 'low' (C2 and D-E).

The minimum set of underlying data of our study called "S1 File" was uploaded. The data will be available in the "Public Repository".

## Statistical analysis

The statistical analysis were conducted in the Software for Statistical and Date Science (STATA), version 13.0 (StataCorp LP®, Texas, USA).

To evaluate the normality, the Kolmogorov-Smirnov test was used and showed absence of normality in the distribution of the variables. Spearman's correlation coefficient was used to verify the degree of statistical dependence between the analyzed variables. The description of the categorical variables was obtained by the distribution of the absolute and relative frequency, total and stratified for the level of PA.

For each estimated density surface, four categories were used, dividing the adolescents into quartiles, based on the density values assigned to each one. Four categories were also used for walkability, dividing individuals into quartiles, based on the walkability values assigned to each one. Thus, all density and walkability categories were modeled using dummies variables, with the reference category being the one with the lowest density (Q1) and walkability (Q1) values.

The chi-square test was used to verify the differences between the kernel density estimate categories and the walkability categories with the level of PA.

The binary logistic regression was used to verify the association between the Kernel density categories of the places for PA, independent variable (classified into quartiles Q1(smallest)—Q4(largest)) and the chance of being "Sufficient PA", dependent variable. Likewise, the association between the walkability categories, independent variable (quartiles Q1 (smallest)—Q4 (largest)) and the chance of being "Sufficient PA", dependent variable, was analyzed.

Among the other variables collected in the study (gender, age, and socioeconomic status (SS)), only SS was analyzed in the statistical modeling as a possible confounding variable. Thus, two logistic regression models were conducted, Model 1 did not include the confounding variable and Model 2 was carried out considering the confounding variable (SS).

The significance of the models was evaluated by the chi-square statistics ($\chi^2$). Was adopted the significance level lower or equal to 0.05.

## Results

Data were collected from 309 adolescents, from these 17 were removed because they lived in census sectors of rural areas. The sample was composed by 292 students (15.38 ± 0.56 years), which 57.19% were female, 52.4% were classified as medium SS, and 52.05% of the sample were classified as "Sufficient PA", according to S3 Table.

The Fig 2 presents the maps with the Kernel density estimative values of the PA places divided into quartiles for radius of 400, 800, 1200 e 1600m. To produce the image, a kernel with cone radius of 400, 800, 1200, and 1600m was placed over each PA place in the data set. Overlapping cones were added to produce a continuous surface, with closer destinations (PA place) producing higher kernel density estimates. It is important to note that kernel density estimates were calculated independently of adolescents' homes. Kernel density values were extracted so that each participant's home location was assigned the kernel density value of the

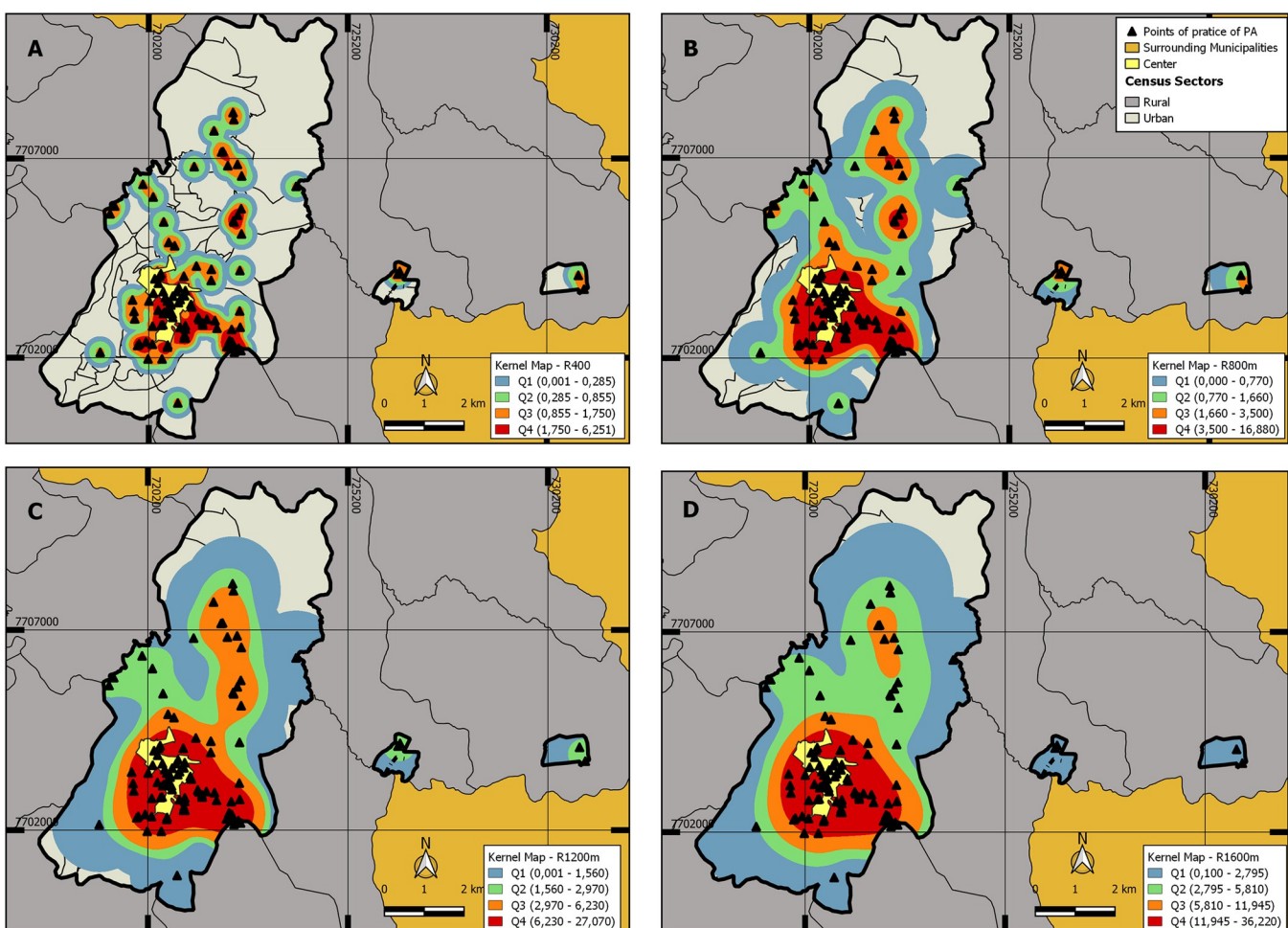

**Fig 2. Kernel density estimative values of the PA places divided into quartiles for radius of 400, 800, 1200 e 1600 meters in Viçosa–MG.** A. Kernel Map—R400m, Kernel Map in Radius of 400 meters. B. Kernel Map—R800m, Kernel Map in Radius of 800 meters. C. Kernel Map—R1200m, Kernel Map in Radius of 1200 meters. D. Kernel Map—R1600m, Kernel Map in Radius of 1600 meters. Q1, quartile 1; Q2, quartile 2; Q3, quartile 3; Q4, quartile 4. Colors of the Quartiles in the Kernel Map: Q1, blue; Q2, green; Q3, orange; Q4, red.

output cell in which they resided. Although estimates are calculated based on the proximity of destinations (PA place) to each other, the values extracted at each home location provide an indication of the proximity and density of destinations in relation to the home location. Thus, the map illustrates that high kernel density estimates indicate high concentration (intensity) of destinations (PA place) indicating greater proximity between the respondent's home and destinations, as observed in Fig 2C and 2D (the map of kernel covers a large part of the study area and with the areas of each quartile delimited, continuously and without irregularities). Low kernel density estimates indicate insignificant and dispersed PA places, as observed in Fig 2A (absence of continuity in the red areas). Finally, moderate kernel density estimates indicate dispersed PA places or occur when an adolescent's residence is located some distance from a set of highly clustered PA place (Fig 2B). The S2 Fig presents the distribution of adolescents in the Kernel Maps, with the density estimative values divided into quartiles for radius of 400, 800, 1200 e 1600m.

The Fig 3 presents the map with the census sectors walkability values divided into quartiles, as well as the distribution of adolescents in the sectors. The quartile 1 represents the census sectors with the lowest values of walkability, while the quartile 4 are the sectors with the best values. It was possible to observe that the regions closer to the city center and in the district of São José do Triunfo the census sectors with the best structure for walkability were located.

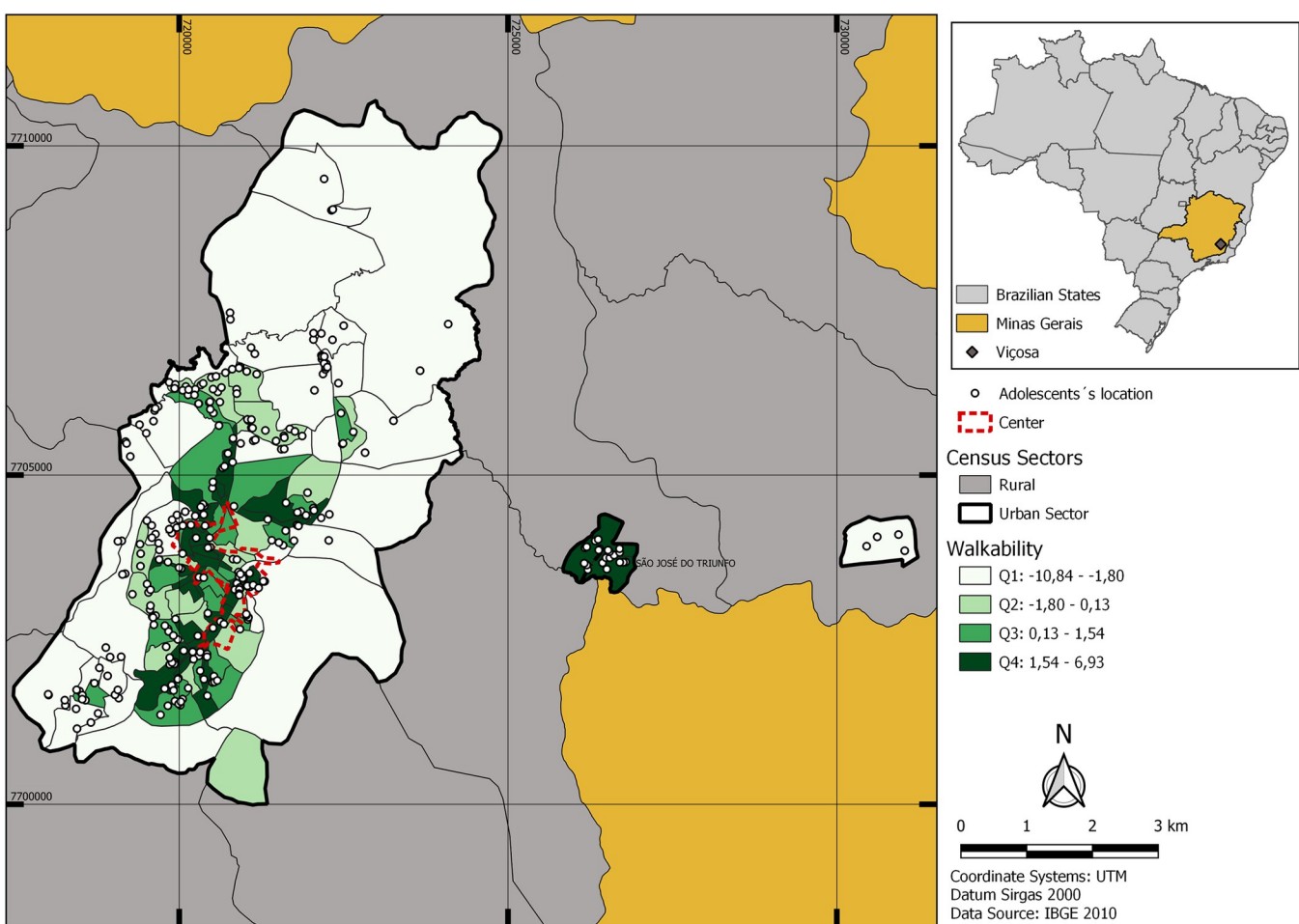

**Fig 3. Map with the census sectors walkability values of Viçosa-MG divided into quartiles, as well as the distribution of adolescents in the sectors.** Q1, quartile 1; Q2, quartile 2; Q3, quartile 3; Q4, quartile 4.

Going beyond, analyzing the Figs 2 and 3 it was noted a pattern of distribution similar in both; the greater intensive (concentration) of places for PA and the best values for walkability were found in the regions closer to the city center. This observation was confirmed carrying out the Spearman (*r*) correlation between density estimate, determined for different radius, and the walkability, with moderate values of correlation ($0.42 \leq r \leq 0.66$).

The Table 1 presents the values of absolute and relative frequency of the levels of PA of the adolescents distributed between the Kernel density quartiles and between the walkability quartiles, as well as the results of the chi-square test to verify if there were significant differences between individuals classified as "Sufficient PA" and "Insufficient PA. The radius of 400m embraced a smaller number of individuals (n = 196), while in the biggest radius all the sample

**Table 1. Values of absolute and relative frequency and the results of the chi-square test of the levels of PA of the adolescents distributed between of the kernel density quartiles and between walkability quartiles.**

| Exposure Variable | Response Category | Distribution of PA locations in each radius | Physical Activity Sufficiency | | |
|---|---|---|---|---|---|
| | | | Total n (%) | Sufficient PA | Insufficient PA |
| | | | 292 (100%) | N (%) | N (%) |
| *Kernel Density*: 400 m | *Total* | 98 | 196 (100%) | 105 (53.57%) | 91 (46.43%) |
| | Quartile 1 | 0 | 49 (25.00%) | 23 (46.94%) | 26 (53.06%) |
| | Quartile 2 | 0 | 49 (25.00%) | 25 (51.02%) | 24 (48.98%) |
| | Quartile 3 | 34 | 48 (24.49%) | 31 (64.58%) | 17 (35.42%) |
| | Quartile 4 | 64 | 50 (25.51%) | 26 (52.00%) | 24 (48.00%) |
| Pearson chi2(3) = 3.385 p = 0.336 | | | | | |
| *Kernel Density*: 800 m | *Total* | 98 | 275 (100%) | 146 (53.09%) | 129 (46.91%) |
| | Quartile 1 | 0 | 68 (24.73%) | 33 (48.53%) | 35 (51.47%) |
| | Quartile 2 | 10 | 69 (25.09%) | 29 (42.03%) | 40 (57.97%) |
| | Quartile 3 | 20 | 69 (25.09%) | 41 (59.42%) | 28 (40.58%) |
| | Quartile 4 | 68 | 69 (25.09%) | 43 (62.32%) | 26 (37.58%) |
| Pearson chi2(3) = 7.428 p = 0.059 | | | | | |
| *Kernel Density*: 1200 m | *Total* | 98 | 291 (100%) | 152 (52.23%) | 139 (47.77%) |
| | Quartile 1 | 3 | 73 (25.09%) | 31 (42.47%) | 42 (57.53%) |
| | Quartile 2 | 12 | 72 (24.74%) | 27 (37.50%) | 45 (62.50%) |
| | Quartile 3 | 18 | 73 (25.09%) | 45 (61.64%) | 28 (38.36%) |
| | Quartile 4 | 65 | 73 (25.09%) | 49 (67.12%) | 24 (32.88%) |
| Pearson chi2(3) = 18.134 p < 0.001* | | | | | |
| *Kernel Density*: 1600 m | *Total* | 98 | 292 (100%) | 152 (52.05%) | 140 (47.95%) |
| | Quartile 1 | 8 | 73 (25.00%) | 25 (34.25%) | 48 (65.75%) |
| | Quartile 2 | 15 | 73 (25.00%) | 31 (42.47%) | 42 (57.53%) |
| | Quartile 3 | 9 | 73 (25.00%) | 48 (65.75%) | 25 (34.25%) |
| | Quartile 4 | 66 | 73 (25.00%) | 48 (65.75%) | 25 (34.25%) |
| Pearson chi2(2) = 9.724 p = 0.008* | | | | | |
| *Walkability* | *Total* | - | 292 (100%) | 152 (52.05%) | 140 (47.95%) |
| | Quartile 1 | - | 75 (25.68%) | 32 (42.67%) | 43 (57.33%) |
| | Quartile 2 | - | 71 (24.32%) | 33 (46.48%) | 38 (53.52%) |
| | Quartile 3 | - | 73 (25.00%) | 39 (53.42%) | 34 (46.58%) |
| | Quartile 4 | - | 73 (25.00%) | 48 (65.75%) | 25 (34.25%) |
| Pearson chi2(3) = 9.077 p = 0.028* | | | | | |

*significative association with physical activity; Total N, total sample; PA, physical activity; %; percentage; m, meters; Sufficient PA, physical activity sufficient; Insufficient PA, physical activity insufficient; p, p-value; Pearson chi2, chi-square.

was embraced (1600m), thus not all residences were attributed density values. As it is a smaller radius, the density calculation did not cover all areas of the urban sector of the municipality. Therefore, this fact does not constitute a loss of data, but a part of the research result. For the 1200 and 1600m radius, all residences had density values assigned.

For all the Kernel densities (in the 400, 800,1200, 1600m radius) analyzed there was a bigger proportion of individuals classified as "Sufficient PA" (~ 52%). Similar proportion of "Sufficient PA" individuals was observed for walkability. The chi-square test showed significant differences between the "Sufficient PA" and "Insufficient PA" groups for both density estimates (p < 0.001) and walkability (p = 0.028). Thus, the quartiles with the highest intensity of PA places (quartiles 3 and 4) contained a significantly greater proportion of those that were "PA Sufficient" at both the 1200 and 1600m radius. Similarly, a greater proportion of individuals who were "PA Sufficient" lived in areas with better walkability values (Quartiles 3 and 4).

Table 2 presents the associations between the PA level of adolescents with Kernel density estimates (radius of 400, 800, 1200 and 1600m) and with the walkability index. The model adjusted for the SS confounding factor showed that this variable was not significant, indicating that the relation between the variables of interest was not significantly affected by SS. Therefore, were presented the results of the analysis of model 1, which was not adjusted for the confounding factor.

Were observed associations between the level of PA and the Kernel density estimative only for the highest intensities at the largest radius places. The increase in the Kernel density estimate for the intensity of the PA places was associated with a probability of being "Sufficient PA" at the Kernel sizes for the radius of 1200 and 1600. The evidence was stronger for quartiles 3 and 4 compared to quartile 1 at 1200m (Q3, OR 2.18 95% CI 1.12–4.22; Q4, OR 2.77 95% CI 1.41–5.43) and 1600m (Q3, OR 3.68 95% CI 1.86–7.30; Q4, OR 3.69 95%CI 1.86–7.30). Furthermore, living in neighborhoods with better walkability was associated with a greater chance of the adolescent being "Sufficient PA". The evidence was stronger for quartile 4 when compared to quartile 1 (Q4, OR 2.58 95% CI 1.33–5.02).

## Discussion

This study provides the strongest evidence to date of the association between the intensity of PA places and PA. The results show that the intensity of PA places is associated with a higher level of PA. At the two largest center distances, individuals in the top two quartiles (with the highest places intensity) were approximately 2.18–3.69 times more likely to be "PA Sufficient." Such results suggest that changes in the distribution of PA places have the potential to increase PA, so that more adolescents are active enough for health.

Since there is little previous research investigating the density estimate the distribution of the places PA in the outcomes of PA, especially in adolescents, it makes it hard to have an opposition of these findings in the existing literature. In general terms, the results found are consistent with studies that examined the density estimate of resources and recreational facilities, stores and commerce in adults [25, 32] and recreational facilities and green spaces in children [28, 29]. The results are also consistent with the findings of a study from Delmenhorst, Germany, in which they analyzed the effect of Kernel density estimates and network distances on the level of PA and observed positive associations between the intensity of destinations with moderate-to-vigorous PA [28].

An important finding of the present study (Figs 2 and 3) was that although density estimates and walkability evaluate the environment differently, both observed that regions closer to the city center had a greater intensity (concentration) of places for AF and the census sectors with better walkability. It is possible that this happened due to the characteristics of the place

**Table 2. Results of the logistic regression between the PA level of adolescents with kernel density estimates (radius of 400, 800, 1200 and 1600m) and with the walkability index.**

| Exposure Variable | Response Category | MVPA | | | | | | MVPA | | | | | |
| --- | --- | --- | --- | --- | --- | --- | --- | --- | --- | --- | --- | --- | --- |
| | | †Model 1 | | | | | | ‡Model 2 | | | | | |
| | | β | SE | OR | 95%CI | p-value | $\chi^2$ | β | SE | OR | 95%CI | p-value | $\chi^2$ |
| *Kernel Density*: 400 m | Quartile 1† | | | | | | | | | | | | |
| | Quartile 2 | 0.16 | 0.40 | 1.18 | 0.53–2.60 | 0.69 | 0.33 | 0.15 | 0.42 | 1.16 | 0.51–2.64 | 0.73 | 0.52 |
| | Quartile 3 | 0.72 | 0.42 | 2.06 | 0.91–4.66 | 0.08 | | 0.72 | 0.42 | 2.05 | 0.90–4.72 | 0.09 | |
| | Quartile 4 | 0.20 | 0.40 | 1.22 | 0.56–2.70 | 0.62 | | 0.14 | 0.42 | 1.14 | 0.50–2.62 | 0.75 | |
| | SS low $^{\phi}$ | | | | | | | | | | | | |
| | SS medium | | | | | | | -0.08 | 0.36 | 0.92 | 0.45–1.89 | 0.82 | |
| | SS high | | | | | | | 0.30 | 0.44 | 1.35 | 0.57–3.18 | 0.49 | |
| *Kernel Density*: 800 m | Quartile 1† | | | | | | | | | | | | |
| | Quartile 2 | -0.26 | 0.34 | 0.77 | 0.39–1.51 | 0.44 | 0.06 | -0.26 | 0.35 | 0.77 | 0.38–1.54 | 0.46 | 0.20 |
| | Quartile 3 | 0.44 | 0.34 | 1.55 | 0.79–3.05 | 0.20 | | 0.42 | 0.36 | 1.53 | 0.79–3.07 | 0.23 | |
| | Quartile 4 | 0.56 | 0.35 | 1.75 | 0.89–3.46 | 0.11 | | 0.52 | 0.36 | 1.68 | 0.83–3.41 | 0.15 | |
| | SS low $^{\phi}$ | | | | | | | | | | | | |
| | SS medium | | | | | | | -0.01 | 0.32 | 0.99 | 0.53–1.86 | 0.97 | |
| | SS high | | | | | | | 0.28 | 0.38 | 1.32 | 0.63–2.79 | 0.47 | |
| *Kernel Density*: 1200 m | Quartile 1† | | | | | | | | | | | | |
| | Quartile 2 | -0.21 | 0.33 | 0.81 | 0.42–1.58 | 0.54 | <0.01 | -0.13 | 0.35 | 0.88 | 0.44–1.75 | 0.72 | <0.01 |
| | Quartile 3 | 0.78 | 0.33 | 2.18 | 1.12–4.22 | 0.02* | | 0.74 | 0.35 | 2.10 | 1.07–4.14 | 0.03* | |
| | Quartile 4 | 1.02 | 0.34 | 2.77 | 1.41–5.43 | <0.01* | | 1.00 | 0.36 | 2.73 | 1.35–5.50 | <0.01* | |
| | SS low $^{\phi}$ | | | | | | | | | | | | |
| | SS medium | | | | | | | -0.04 | 0.33 | 0.96 | 0.51–1.82 | 0.89 | |
| | SS high | | | | | | | 0.33 | 0.38 | 1.39 | 0.66–2.94 | 0.39 | |
| *Kernel Density*: 1600 m | Quartile 1† | | | | | | | | | | | | |
| | Quartile 2 | 0.35 | 0.34 | 1.42 | 0.73–2.77 | 0.31 | <0.01 | 0.26 | 0.36 | 1.30 | 0.65–2.60 | 0.46 | <0.01 |
| | Quartile 3 | 1.30 | 0.35 | 3.68 | 1.86–7.30 | <0.01* | | 1.20 | 0.36 | 3.32 | 1.65–6.69 | <0.01* | |
| | Quartile 4 | 1.30 | 0.35 | 3.69 | 1.86–7.30 | <0.01* | | 1.22 | 0.37 | 3.39 | 1.63–7.03 | <0.01* | |
| | SS low $^{\phi}$ | | | | | | | | | | | | |
| | SS medium | | | | | | | -0.04 | 0.33 | 0.96 | 0.50–1.85 | 0.91 | |
| | SS high | | | | | | | 0.37 | 0.39 | 1.45 | 0.67–3.13 | 0.34 | |
| Walkability | Quartile 1† | | | | | | | | | | | | |
| | Quartile 2 | 0.15 | 0.33 | 1.17 | 0.61–2.24 | 0.49 | 0.03 | 0.15 | 0.34 | 1.16 | 0.59–2.28 | 0.66 | 0.03 |
| | Quartile 3 | 0.43 | 0.33 | 1.54 | 0.81–2.95 | 0.78 | | 0.41 | 0.34 | 1.51 | 0.77–2.94 | 0.22 | |
| | Quartile 4 | 0.95 | 0.34 | 2.58 | 1.33–5.02 | <0.01* | | 0.95 | 0.35 | 2.58 | 1.30–5.12 | <0.01* | |
| | SS low $^{\phi}$ | | | | | | | | | | | | |
| | SS medium | | | | | | | -0.15 | 0.31 | 0.86 | 0.47–1.59 | 0.64 | |
| | SS high | | | | | | | 0.26 | 0.37 | 1.30 | 0.63–2.68 | 0.48 | |

*significative association with physical activity; †, reference category, $\phi$, reference category; MVPA, moderate to vigorous physical activity; β, beta coefficient; SE, standard error; OR, odds ratio; 95%CI, confidence interval of 95%; $\chi^2$, chi-square; m, meters; SS, socioeconomic status.

†Model 1: Model without adjustment.

‡Model 2: Model adjusted by SS.

where the study was carried out, as it is a small city, where usually the process of development occurs from the central region to the periphery [52], as observed in Viçosa, where in the regions closer do the center there are concentrated areas for practicing PA and the best

structures for walking, which can pose problems for other people living in more peripheral regions, who suffer from the lack of adequate facilities for moving around and performing PA.

It is important to highlight that 52% of the adolescents were classified as "Sufficient PA", both in density and walkability analyses. However, a high proportion (48%) were below the values considered appropriate [37]. Previous studies indicated that a higher availability and access to places for PA [53–55] and neighborhoods with better walkability [56, 57] were associated with a greater fulfilment of the guidelines of MVPA by the adolescents. All these findings confirm the need for better city planning, which should provide a greater availability of different places for PA, a more intense distribution, as well as better street structures for walking, in order to encourage changes in people's behaviors.

The results show a greater proportion of "Sufficient PA" individuals in the quartiles with the highest intensity of PA places (quartiles 3 and 4) in the 1200m and 1600m radius. Consistent results were observed when analyzing the density of stores and business establishments in the PA of adults, with a bigger proportion of sufficiently active individuals in the quintiles with the bigger radius (800 e 1200m) [25]. Despite the difference in age range between the studies, a greater intensity (concentration) of destinations was observed in the largest radius sizes. This means that larger radius covers a larger area, and thus can concentrate more destinations and to generate greater densities. The presence of intensity of destinations in larger radius may help explain the difference in PA levels among adolescents, due to the need to travel greater distances from their homes to destinations (for example, PA places).

The intensity of the PA places was associated with the PA level at radius of 1200 and 1600 m. Specifically, as the intensity of places increased, were more likely to be "Sufficient PA", this being significant at both 1200 m and 1600 m for quartiles 3 and 4 (the quartiles with the most intensely distributed PA sites). King et al. [25] also observed positive associations for the kernel density estimate of 800 and 1200 m between the quintiles 1 and 3, quintiles 1 and 4 and quintiles 1 and 5. In making sense of the results, higher values of the Kernel density estimate indicate a greater concentration of PA places (Quartiles 3 and 4), and this effect can provide increase in PA. This way, a grouping of places to practice PA means greater availability for use by a greater number of people and consequently influence their PA levels. It is true that in general people seek to do PA in places close to their homes; however, since the concentration of PA locations only occurred at larger radius, part of the PA level can also be explained by the displacement of individuals to the PA places, contributing to a longer time in MVPA.

The association between the density places of PA and PA at 1200 e 1600m in our analysis is noteworthy. Although intensely distributed (concentrated) places at 400 and 800 m may still encourage PA, this may be insufficient to exert health effects, while 1200 and 1600m may be more appropriate distances. To achieve PA sufficiency and receive health benefits, adolescents need to be active more frequently, for longer periods of time, which can be achieved through active travel to PA places located at a greater distance of their residences. It is also important to reflect from the perspective of the rational use of resources, as the costs of building and maintaining infrastructure for PA are high, so there are substantial cost reduction benefits if PA places are located within 1200 m and 1600 m from most houses, in order to provide access to the entire population.

The results showed positive associations between the walkability and the level of PA, where the individuals that lived in best census sectors had a bigger chance of being "Sufficient PA" compared with the ones that lived in sectors with smaller values of walkability. Consistently, Hinckson et al. [56] found positive associations between MVPA and the walkability, with estimated differences of 8 minutes more of MVPA per day of the individuals from the quartile 3 (bigger walkability) when compared to those from quartile 1.

Sallis et al. [57] also observed associations between neighborhoods with greater capacity of walking compared to the neighborhood with fewer capacity of walking. Another interesting finding from this study was that adolescents who live in walkable neighborhoods reported less television time, time in the car, and total sitting time. All these findings reinforce the need for neighborhoods with better environmental attributes, so that they can encourage walking and other types of active displacement among adolescents, such as cycling and skateboarding, and consequently promote healthier behaviors, such as to increase PA and reduce sedentary behavior.

This study presents some limitations. First of all, the data regarding the census sectors were relative to the year of 2010, and may not reflect environmental changes that may have taken place in these areas over this time. However, it was something that we were not able to control since there we no updates of the data by IBGE. Finally, the exclusion of the students that live in the census sectors of the rural areas, because of lack of data of these regions.

As a strong point, the present analysis using Kernel density estimative of the distribution of AF places represents an important advance in the study of the relationship between destinations (places) and PA. Kernel density estimative expresses the distance and density of destinations (places) [22]. By using Kernel density estimative, this study was able to weight and classify access to PA places. Another important consideration was the use of four different sizes of Kernel, because it allowed the comparison of the effects of different distances in the level of PA. Another strong point was the use of objective measures for the evaluation of the PA and for the evaluation of the built environment, since many studies use subjective measures to capture both informations, due to the practicality of the application and collection of the informations. It is worth mentioning that, as far as we know, this is the first Brazilian study using the Kernel density estimate to verify the influence of the density of the places for PA in the active behavior of the adolescents. Finally, it is worth stressing the realization of this study in a small city, since the studies with these characteristics usually occur in large urban centers.

## Conclusion

The observed association between the distribution of PA locations and PA at 1200 and 1600m is consistent with our hypothesis that more intensely distributed destinations would be associated with a greater chance of being "Sufficient PA". As well as living in places with better walkability structure can increase the likelihood of adolescents being "Sufficient PA".

The results suggest that increasing the intensity of destinations in areas where they are more dispersed; and planning neighborhoods with greater walkability can increase the likelihood of adolescents being "Sufficient PA".

These findings have important policy implications, both in relation to the better distribution of places for PA in the cities and, also improvements in the physical infrastructures related to the walkability in the neighborhoods, in order to represent a bigger effect of mobility in peoples lives and contribute to the increase of PA.

## Supporting information

**S1 Fig. Visualization the study area, Viçosa–MG.** UTM; Universal Transverse Mercator System.
(TIF)

**S2 Fig. Distribution of the adolescents in the kernel maps, with the density estimative values divided into quartiles for radius of 400, 800, 1200 e 1600 meters in Viçosa–MG.** Fig 2A. Kernel Map—R400m, Kernel Map in Radius of 400 meters. Fig 2B. Kernel Map—R800m,

Kernel Map in Radius of 800 meters. Fig 2C. Kernel Map—R1200m, Kernel Map in Radius of 1200 meters. Fig 2D. Kernel Map—R1600m, Kernel Map in Radius of 1600 meters. Q1, quartile 1; Q2, quartile 2; Q3, quartile 3; Q4, quartile 4; km, kilometer. Coloring of Quartiles in the Kernel Map: Q1, blue; Q2, green; Q3, orange; Q4, red.
(TIF)

**S1 Table. Descriptive measures of the variables related to the walkability index.** un/km, unit per kilometer; %, percentage; SD, standard deviation; IQ, interquartile interval.
(DOCX)

**S2 Table. Descriptive measures of the kernel density estimative in the radius of 400m, 800m, 1200m e 1600m.** SD, standard deviation; IQ, interquartile range.
(DOCX)

**S3 Table. Average values and interquartile interval of the continuous variables and values of the absolute and relative frequency of the categorical variables, for the total sample and separated by sex.** *statistical significance; †, reference category; SD, standard deviation; LPA, light physical activity; MVPA, moderate to vigorous physical activity; SS, socioeconomic status.
(DOCX)

**S1 File. Code, Participant code; STEPS, Steps number per day a; LPA_Cont, Light Physical Activity in minutes; LPA_Cat, Categorical variables for light physical activity; MPVA, Moderate to vigorous physical activity in minutes; MVPA_Cat, Categorical variables for minutes of moderate to vigorous physical activity;; SS, Socieconomic Status; KDE_ 400, Continuous values for KDE for the 400m radius; KDE_ 800, Continuous values for KDE for the 800m radius; KDE_ 1200, Continuous values for KDE for the 1200m radius; KDE_ 1600, Continuous values for KDE for the 1600m radius; Res_Density, Residential Density Class_Quartile_400, 800, 1200, 1600, Division of adolescents into quartiles according to the KDE for the 400m, 800m, 1200m, 1600m radius; Res_Density, Residential Density in census sectors; Com_Density, Commercial Density in census sectors; Conect_Density, Street Connectivity Density in census sectors; Percent_Sidewalks, Percentage of sidewalks in census sectors; Percent_Light, Percentage of public lighting in census sectors; Walk_index, Walkability Index; Quartile_Walk_index Division of adolescents into quartiles according to the Walkability Index.**
(XLS)

## Acknowledgments

The authors would like to thank all the students and their parents or legal guardians who participated in the study and the teachers, educators and principals who facilitated the research to take place.

## Author Contributions

**Conceptualization:** Isabella Toledo Caetano, Rogério César Fermino, Renato de Oliveira Falcão, Paulo Roberto dos Santos Amorim.

**Formal analysis:** Isabella Toledo Caetano, Renato de Oliveira Falcão.

**Investigation:** Isabella Toledo Caetano.

**Methodology:** Isabella Toledo Caetano, Renato de Oliveira Falcão, Paulo Roberto dos Santos Amorim.

**Resources:** Rogério César Fermino.

**Supervision:** Isabella Toledo Caetano, Rogério César Fermino, Paulo Roberto dos Santos Amorim.

**Writing – original draft:** Isabella Toledo Caetano, Renato de Oliveira Falcão, Paulo Roberto dos Santos Amorim.

**Writing – review & editing:** Isabella Toledo Caetano, Rogério César Fermino, Renato de Oliveira Falcão, Paulo Roberto dos Santos Amorim.

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
