## [Decision Letter · Decision Letter 0]

18 Sep 2023

PONE-D-23-15455Built environment and Physical Activity in adolescents: use of the Kernel Density Estimation and the Walkability IndexPLOS ONE

Dear Dr. Caetano,

Thank you for submitting your manuscript to PLOS ONE. After careful consideration, we feel that it has merit but does not fully meet PLOS ONE’s publication criteria as it currently stands. Therefore, we invite you to submit a revised version of the manuscript that addresses the points raised during the review process.

Following a comprehensive evaluation by the expert reviewer and my own assessment, it has become evident that your manuscript requires revisions before it can be considered for publication. I have outlined some points for your attention at the end of this message, which complement the issues raised by the reviewer. Please review my comments and those of the reviewer. We acknowledge that these revisions may demand considerable effort, but we firmly believe that, with your commitment, the manuscript can meet the necessary standards for publication in PloS One.==============================

We look forward to receiving your revised manuscript.

Kind regards,

Vinícius Silva Belo

Academic Editor

PLOS ONE

Journal Requirements:

4. Please upload a new copy of Figure 1as the detail is not clear. Please follow the link for more information: " ext-link-type="uri" xlink:type="simple">https://blogs.plos.org/plos/2019/06/looking-good-tips-for-creating-your-plos-figures-graphics/"
https://blogs.plos.org/plos/2019/06/looking-good-tips-for-creating-your-plos-figures-graphics/

5. We note that Figures 2 and 3 in your submission contain [map/satellite] images which may be copyrighted. All PLOS content is published under the Creative Commons Attribution License (CC BY 4.0), which means that the manuscript, images, and Supporting Information files will be freely available online, and any third party is permitted to access, download, copy, distribute, and use these materials in any way, even commercially, with proper attribution. For these reasons, we cannot publish previously copyrighted maps or satellite images created using proprietary data, such as Google software (Google Maps, Street View, and Earth). For more information, see our copyright guidelines: http://journals.plos.org/plosone/s/licenses-and-copyright.

          1.You may seek permission from the original copyright holder of Figures  2 and 3 to publish the content specifically under the CC BY 4.0 license. 

Additional Editor Comments:

The language and writing style in your manuscript appear to reflect characteristics of Brazilian Portuguese, which may not be suitable for an English-language scientific publication. It is imperative that you seek the assistance of a professional with expertise in English-language scientific writing to comprehensively review and revise the entire article. This step is vital to ensure clarity, coherence, and adherence to the standards of English scientific writing.

The reviewer and I have noted difficulties in comprehending both the manuscript in general, especially concerning issues related to KDE, acronyms, variables, quartiles, and more. Therefore, the enhancement of writing should extend to ensuring greater clarity throughout the entire article, including abstract, tables and figures.

Please discuss the statistical power for each analysis based on the effectively studied sample size. Additionally, explain why a specific school was not included and how this exclusion might affect the results. Clarify the random participant selection process. Please, also reconsider the sample size formula adopted, as it is not adequate for analytical studies.

Although Pearson correlation was used, there is no discussion about the data distribution. Is there a normal distribution?

Ensure the presentation of categories for qualitative variables in the statistical analysis section.

Detailed explanations of the logistic regression models and their results are lacking. Please refer to the STROBE checklist for guidance and correct these deficiencies throughout the entire article. Provide information on model adjustments and residual analyses. Clarify which variables were modeled, the criteria for their selection, and the methods employed for confounding control.

Considering the presence of quantitative data, it is relevant to include other multiple regression analytical procedures, particularly those treating the outcome as quantitative. This could enhance the potential for establishing causality and improve the validity of the results.

Also conduct an analysis of data loss and discuss its potential impact on the associations presented in your study.

Finally, to aid reader comprehension of your results, please provide one or more detailed maps of the study area. Additionally, insert explanatory images, such as those from Google tools.

Reviewers' comments:

Reviewer's Responses to Questions

**Comments to the Author**

1. Is the manuscript technically sound, and do the data support the conclusions?

Reviewer #1: Yes

2. Has the statistical analysis been performed appropriately and rigorously? 

Reviewer #1: Yes

3. Have the authors made all data underlying the findings in their manuscript fully available?

Reviewer #1: Yes

4. Is the manuscript presented in an intelligible fashion and written in standard English?

Reviewer #1: Yes

5. Review Comments to the Author

Reviewer #1: Introduction:

The study at hand is notable for its comprehensive compilation of evidence. However, it is suggested that the authors consider a more direct and efficient approach, as the paper's current readability leaves room for improvement. The reader's experience can be enhanced by providing a clear direction, such as addressing questions like:

Why is research examining the distribution density of places for physical activity (PA) significant in daily academic and professional practices?

What is the concept of "walkability" in straightforward terms?

How do these concepts lead to hypotheses regarding the complex associations between the distribution density of places for PA, walkability, and PA levels?

Methods:

A noteworthy point arises in lines 186 to 200 where the concept discussed could be better placed in the introduction. This concept is not merely a variable but a way to perceive streets, squares, parks, avenues, and more.

Results (general comments):

Regarding my knowledge of physical activity classifications and concepts, it is crucial to avoid categorizing adolescents as "inactive." None are entirely devoid of physical activity; rather, their level of physical movement relates to health indicators. This implies that the classification should indicate whether the total physical activity was sufficient or insufficient for health. There is evidence suggesting a need for greater precision in these conceptual definitions.

The tables provided appear rather convoluted. Given my prior experience in writing articles on environmental factors and physical activity, it is advisable, in my opinion, to enhance the article's clarity to attract readers of all backgrounds. To achieve this, it is recommended to present the tables with:

A description of the variables

Figures illustrating the assessment method and results

The proposed associations as outlined in the study's objectives

There seems to be no justification for an excessive number of tables in a study focusing on the association between just three variables, even if they are latent variables.

Lines 414-426 contain arguably the most significant study results. While the text provides a wealth of information, it is essential to address how these findings translate to the real world. Do increased distribution densities of places for PA enhance walkability? Does enhanced walkability correlate with increased physical activity among Brazilian adolescents? Is walkability a mediator or moderator in the positive relationship between the distribution density of places for PA and physical activity levels? The description of results in the tables should aim to clarify these questions.

Discussion:

To illustrate, I will cite one example, but this issue persists throughout the entire discussion: "The lack of association for the radius of 400 and 800 meters can be explained by the fact that the density produced for these smaller radii can produce estimations with a lot of peaks and discontinuous surfaces depending on the quantity of points observed [50], different from what occurs in the larger radius (lines 513-517)." The discussion section, in addition to being excessively lengthy, is riddled with justifications for unexpected results. The authors frequently attempt to align their study's results with studies that had different objectives, resulting in a confusing and tedious read. It is challenging to discern whether there is a positive, negative, or no association between the analyzed variables up to line 517. The authors appear fixated on discussing descriptive results, which are abundantly available in numerous scientific articles. While it is well-known that many adolescents have insufficient levels of physical activity, this study delves into a complex association, primarily explained numerically, as suggested in the quoted passage. In my academic opinion, the study's strength could lie in its discussion, bridging these intriguing variables with real-world applications, offering valuable insights to readers, especially myself. Unfortunately, the discussion, as currently presented, falls short of these expectations.

Conclusion:

Lines 578-583: In my understanding, the conclusion should translate the study's results into a practical and straightforward perspective. It serves as the starting point for a genuine discussion of the study's impact on people's lifestyles.

6. PLOS authors have the option to publish the peer review history of their article (what does this mean?). If published, this will include your full peer review and any attached files.

Reviewer #1: **Yes: **Vanilson Batista Lemes

---

## [Author Response · Author response to Decision Letter 0]

5 Feb 2024

Viçosa, Minas Gerais, Brazil. February 05th 2024.

Dear Vinícius Silva Belo, Academic Editor, PLOS ONE Initially, we would like to thank the editor and reviewer for their contribution to the manuscript entitled “Built environment and Physical Activity in adolescents: use of the Kernel Density Estimation and the Walkability Index”, as well as for the time spent on these contributions. We considered that the comments were extremely important and constructive. They contributed to improve the quality of the manuscript. It is important to highlight that, for some of the contributions, we tried to better explain our opinions regarding the subject.

We describe each one of the comments along with our answers, which are highlighted in yellow. The insertions in the article are also highlighted in yellow.

 Hoping to meet the quality requirements of PLOS ONE, we are at your disposal for whatever is necessary.

Kind regards.

Journal Requirements:

R. A review was carried out on the manuscript to verify the PLOS ONE style, so modifications were made to the nomenclature referring to the database, which was replaced by “Supporting Information Files”. 

R. Along with the revised manuscript, the minimum set of underlying data of our study called “Supporting Information Files” will be uploaded. Furthermore, the data will be available in the “Public Repository” (https://data.mendeley.com/datasets/2w87n5fznt/3).

R. Information regarding the guidelines followed by the study and the approval of the Ethics and Research Committee involving human beings at the Federal University of Viçosa was already included in the manuscript, in the “Methods” section, subsection ‘Declaration of Ethics’ (Page 7, Lines 157 - 159).

To complement this section, information was added regarding the terms of consent signed by the students and their legal guardians. (Page 7, Lines 160 - 162).

“The students had to present an Informed Consent Form and an Informed Assent Term duly signed by their legal guardians and by the adolescents to participate in the study.”

4. Please upload a new copy of Figure 1as the detail is not clear. Please follow the link for more information: https://blogs.plos.org/plos/2019/06/looking-good-tips-for-creating-your-plos-figures-graphics/" https://blogs.plos.org/plos/2019/06/looking-good-tips-for-creating-your-plos-figures-graphics/

R. A new copy of Figure 1 was inserted to make the details of the image clearer.

5. We note that Figures 2 and 3 in your submission contain [map/satellite] images which may be copyrighted. All PLOS content is published under the Creative Commons Attribution License (CC BY 4.0), which means that the manuscript, images, and Supporting Information files will be freely available online, and any third party is permitted to access, download, copy, distribute, and use these materials in any way, even commercially, with proper attribution. For these reasons, we cannot publish previously copyrighted maps or satellite images created using proprietary data, such as Google software (Google Maps, Street View, and Earth). For more information, see our copyright guidelines: http://journals.plos.org/plosone/s/licenses-and-copyright.

 1.You may seek permission from the original copyright holder of Figures 2 and 3 to publish the content specifically under the CC BY 4.0 license. 

R. It is important to clarify that Google software (Google Maps, Street View and Earth) were not used to create the maps present in Figures 2 and 3. These tools were used prior to the preparation of these figures, only to locate the residence of the adolescents and spaces for practicing physical activity and which were later converted into geographic coordinates. Therefore, in Figures 2 and 3 there is no satellite image or map from Google tools.

The maps present in Figures 2 and 3 were created using the ShapeFiles files that contain the division of the Brazilian territory into municipalities and the division of municipalities into census sectors. These files, in turn, are made available free of charge by the Brazilian Institute of Geography and Statistics – IBGE, a public body linked to the Brazilian Federal Government whose main functions are the coordination, consolidation and availability of geographic data from the entire Brazilian territory.

Specifically, the set of Shapefile files, used to create Figures 2 and 3, relating to the territorial division of Brazilian municipalities is available at the following address: https://www.ibge.gov.br/geociencias/organizacao-do-territorio/malhas-territoriais/15774-malhas.html

In turn, the set of files referring to the census sectors, also used in the creation of Figures 2 and 3, are available at the following address:

https://www.ibge.gov.br/geociencias/organizacao-do-territorio/malhas-territoriais/26565-malhas-de-setores-censitarios-divisoes-intramunicipais.html?edicao=30113t=downloads.

 It is also necessary to clarify that Figures 2 and 3 constitute original results of the analyzes carried out and, therefore, are authored by the present authors of this article and that by submitting it they authorize the publication of these Figures.

R. The caption referring to the information from the Excel database was inserted into the manuscript, in the “Supporting Information” section (Pages 30 and 31, Lines 645 - 666). 

Additional Editor Comments:

1. The language and writing style in your manuscript appear to reflect characteristics of Brazilian Portuguese, which may not be suitable for an English-language scientific publication. It is imperative that you seek the assistance of a professional with expertise in English-language scientific writing to comprehensively review and revise the entire article. This step is vital to ensure clarity, coherence, and adherence to the standards of English scientific writing.

R. A careful review of scientific writing, in English, was carried out on the entire manuscript.

2. The reviewer and I have noted difficulties in comprehending both the manuscript in general, especially concerning issues related to KDE, acronyms, variables, quartiles, and more. Therefore, the enhancement of writing should extend to ensuring greater clarity throughout the entire article, including abstract, tables and figures.

R. We agree with you that some parts of the manuscript are confusing when referring to KDE, acronyms, variables, quartiles. To resolve this problem, we carried out a careful review of the entire manuscript, where we rewrote the summary, tried to make the information regarding Kernel Density Estimate (KDE) and walkability clearer in the introduction, in the methods section we added information about the variables and their classification categories (quartiles) and improving of the information contained in tables and figures. Furthermore, we replaced the acronym KDE with "Kernel density estimative" or "density estimative" to make the reading clearer. All these changes are highlighted in yellow in the text and can be found on the following pages:

• Abstract – (Page 2, Lines 32 – 55)

• Introduction 

- KDE information (Page 4, Lines 82 – 83; Page 4, Lines 85 - 89; Pages 4 and 5, Lines 96 - 107)

- Walkability information (Page 3, Lines 69 – 71; Page 3, Lines 73 - 79)

- Hypothesis (Pages 6 and 7, Lines 127 - 133)

- Objective (Page 6, Lines 134 - 138)

• Methods 

- Walkability variables (Pages 8 and 9, Lines 199 – 208; Page 9, Lines 232 – 234)

- Walkability categories (Page 9, Lines 236 – 237)

- KDE variables (Page 10, Lines 244 - 252; Page 11, Lines 253 – 268; Pages 12 and 13, Lines 297 - 312)

- KDE categories (Page 12, Lines 290 – 291; Page 13, Lines 320 - 325)

• Figures 

- Figure 1 (Page 12, Line 283)

- Figure 2 (Pages 16 and 17, Lines 389 - 409; Page 17, Lines 411 - 417)

- Figure 3 (Page 17, Lines 419 – 420; Page 17, 426 - 428)

- Supplementary Figure 1 (Page 7, Lines 173)

- Supplementary Figure 2 (Page 17, Lines 407 – 409; Pages 28 and 29, Line 616 - 624)

• Tables

- Table 1 (Page 18, Lines 436 – 454; Page 19 and 20)

- Table 2 (Page 21, Lines 466 – 481; Pages 22 and 23)

- Supplementary Table 1 (Page 29, Lines 626 - 629)

- Supplementary Table 2 (Page 13, Lines 317 – 319; Page 29, Line 631 - 633)

3. Please discuss the statistical power for each analysis based on the effectively studied sample size. Additionally, explain why a specific school was not included and how this exclusion might affect the results. Clarify the random participant selection process. Please, also reconsider the sample size formula adopted, as it is not adequate for analytical studies.

R. It is important to highlight that we followed the editor's guidelines regarding the formula used for analytical studies. Initially, we had estimated the sample size calculation using the criteria described in the text and observed that a minimum sample of 305 individuals would be required. We added information on the sample size, calculated a posteriori, using the G*Power software in accordance with the procedures for determining the sample size for Logistic Regression.

 Clarify that all statistical analyzes were carried out with a sample size of 292 students. The manuscript states that we collected information from 309 students, a number exceeding the minimum sample (N = 305) established by the sample calculation. However, when we carried out the initial analyses, 17 students did not have their information analyzed because they lived in census tracts in rural areas and, therefore, were removed from the study. Thus, the study sample in which all analyzes were carried out was 292 students.

Regarding the school that was not included, the city at the time of the study had seven public schools that offered secondary education. All seven schools were invited to participate in the research, however, one school declined the invitation. We believe that as the refusal to participate occurred before we started collecting the data, this fact was minimized by the researchers, increasing the number of participants in other schools and avoiding future losses in the study.

The information described in the text was incomplete and it was not clear how the random selection process of participants occurred. The selection took place through a draw, based on the list of enrolled students.

We sought to resolve all the issues raised in the manuscript, by rewriting how the sample calculation was carried out (Page 6, Lines 146 – 155).

“For sample size estimation it was considered the established population of 968 (number of students enrolled in the first year of high school, in 7 public schools of the city), the estimated prevalence for PA practice at recommended levels of 50% [37], drawing effect of 1.1, interval 95% confidence. Based on these criteria, a minimum sample size of 305 adolescents was reached. The sample size, calculated a posteriori, allows detecting associations with an odds ratio greater than 2.3 with a power minimum of 87% for an alpha value of 5%. For this purpose, the G*Power software version 3.1.9.7 was used. The final sample consisted of 309 students, belonging to six public schools (five state schools and one federal). To obtain the representative sample, the students from each school were selected by drawing lots, based on the list of enrolled students [38].” 

4. Although Pearson correlation was used, there is no discussion about the data distribution. Is there a normal distribution?

R. Information regarding data distribution was not actually presented. However, we had carried out the analysis and the data showed a lack of normality in the distribution. Therefore, we realized that there was a mistake on our part and the correlation analyzes were redone using the appropriate “Spearman Correlation” analysis.

It is important to highlight that when performing the Spearman Correlation analysis, the results were consistent with those found previously, with values even higher than those found before. Spearman's correlation (r) showed moderate correlation values (0.42 ≤ r ≤ 0.66) between KDE, determined for the different radii, and walkability,

Missing information was presented in the statistical analysis section (Page 15; Lines 357 - 360) and changes regarding the results of the correlation analysis in the results section (Pages 17 and 18; Lines 432 - 435).

“To evaluate the normality, the Kolmogorov-Smirnov test was used and showed absence of normality in the distribution of the variables. Spearman’s correlation coefficient was used to verify the degree of statistical dependence between the analyzed variables.” (Page 15; Lines 357 - 360).

“This observation was confirmed carrying out the Spearman (r) correlation between density estimate, determined for different radius, and the walkability, with moderate values of correlation (0.42 ≤ r ≤ 0.66).” (Pages 17 - 18; Lines 432 - 435).

5. Ensure the presentation of categories for qualitative variables in the statistical analysis section.

R. Information regarding qualitative variables was inserted in the “Statistical analysis” section (Page 14, Lines 362 – 367).

For each estimated density surface, four categories were used, dividing the adolescents into quartiles, based on the density values assigned to each one. Four categories were also used for walkability, dividing individuals into quartiles, based on the walkability values assigned to each one. Thus, all density and walkability categories were modeled using dummies variables, with the reference category being the one with the lowest density (Q1) and walkability (Q1) values. 

6. Detailed explanations of the logistic regression models and their results are lacking. Please refer to the STROBE checklist for guidance and correct these deficiencies throughout the entire article. 

R. As suggested by the editor, detailed information about the logistic regression models were added to the manuscript in the “Statistical Analysis” section with the following information (Pages 15 and 16, Lines 370 - 380).

“The binary logistic regression was used to verify the association between the Kernel density categories of the places for PA, independent variable (classified into quartiles Q1(smallest) - Q4(largest)) and the chance of being "Sufficient PA", dependent variable. Likewise, the association between the walkability categories, independent variable (quartiles Q1 (smallest) - Q4 (largest)) and the chance of being "Sufficient PA", dependent variable, was analyzed.

Among the other variables collected in the study (gender, age, and socioeconomic status (SS)), only SS was analyzed in the statistical modeling as a possible confounding variable. Thus, two logistic regression models were conducted, Model 1 did not include the confounding variable and Model 2 was carried out considering the confounding variable (SS).”

 Furthermore, in the “Results” section we sought to clarify the information in the table and the description of the results regarding the results of the Logistic Regression (Page 21, Lines 466 – 481; Pages 22 and 23).

7. Provide information on model adjustments and residual analyses. 

R. Regarding model adjustment and residual analysis (error terms), these analyzes were not used in the present study. For the adjustment check of the model, the adjustment coefficient R2 is used. However, according to Favero and Belfiore (2017), in logistic regression models there is no adjustment coefficient R2 as in traditional models estimated by Ordinary Least Squares. As the dependent variable is a qualitative variable, it makes no sense to discuss what percentage of its variance can be explained by the variance of the predictor variables. Still according to the author, many researchers present in their studies the coefficient called McFadden's pseudo R2, however, the use of this coefficient is restricted to cases in which they are interested in comparing two or more different models, which did not happen in the present study.

Regarding the analysis of residues, Favero and Belfiore (2017) states that Binary Logistic Regression has its techniques elaborated based on maximum likelihood estimation, which has the function of finding parameters that maximize the probability of occurrence of an event, whose dependent variable is presented in a qualitatively dichotomous way and, unlike Ordinary Least Squares estimation methods, there is no way to minimize the sum of the squared residuals without there being arbitrary weighting. Therefore, in Binary Logistic Regression models, residual analysis does not impact the estimated parameters or the overall significance of the model.

Source: Favero, Luiz Paulo e Belfiore, Patrícia: Manual de Análise de Dados, 1 ed., Rio de Janeiro: Elsevier, 2017.

8. Clarify which variables were modeled, the criteria for their selection, and the methods employed for confounding control.

R. In responding to the editor's comment above (6. Detailed explanations about logistic regression models are missing...), we explained how the variables were modeled and the methods used to control confounding in the “Statistical Analysis” section (Pages 15 and 16, Lines 370 - 380).

As for the information regarding the “criteria for selecting variables” in the manuscript, the “Obtaining and Preparing Variables” section presents the sources for capturing the information (Pages 7 and 8, Line 179). In the section “Related Information to Walkability” (Pages 8 and 9, Lines 199) the entire process of capturing and analyzing the variables used to create the walkability index is explained. Regarding KDE, in the section “Places for PA used to estimate Kernel Density Estimate” (Page 10, Lines 244 – 252; Page 11, Lines 253 - 268) the entire process of capturing and analyzing the variables used for KDE estimation is described.

Regarding the use of possible confounding variables in statistical modeling, it is important to clarify that among the other variables collected in the study (gender, age, and socioeconomic status (SS)), the only one that made sense for us to test as a possible confounding variable was the SS. This is because it is a variable that would be related to both of the independent variable (KDE and walkability categories) and the dependent variable (AF categories). Thus, by using this variable we sought to obtain more precise results that would indicate another possible important association with the level of PA. However, there was no association, indicating that the relationship between the variables of interest was not significantly affected by SS. Therefore, it was decided to discuss the results of the analysis without the confounding variable in the statistical modeling.

Regarding the variables gender and age, these could not be considered as confounding variables. To be considered a confounding variable, it must be related to the dependent and independent variables analyzed. In the present study, the variables of gender and age can only influence the level of PA (dependent variable), which does not occur in relation to variables related to the environment (KDE and Walkability).

It is important to highlight that as the explanation of the SS confounding variable was added to the statistical analyzes (Pages 15 and 16, Lines 370 - 380) and in Table 2 (Results section) (Pages 22 and 23), the way in which this variable was collected was inserted in the “Method” section (Page 14, Lines 348 - 352).

“Confounding Variable

The socioeconomic status (SS) of the family was extracted from the Economic Classification Criteria of the Brazilian Association of Research Companies [42]. According to the final score, the participants’ SS were classified into 3 classes: ‘high’ (classes A and B1), ‘average’ (B2 and C1) and ‘low’ (C2 and D-E).” (Page 14, Lines 348 - 352).

9. Considering the presence of quantitative data, it is relevant to include other multiple regression analytical procedures, particularly those treating the outcome as quantitative. This could enhance the potential for establishing causality and improve the validity of the results.

R. During the process of analysis, the possibility of including Multiple Regression models using quantitative data was studied, in particular the Linear Regression Model, with estimation by Ordinary Least Squares (OLS) given that its estimators are Best Unbiased Linear Estimators (MELVN) (Woldrige, 2009).

However, for the purpose of the research, the interpretation of the results generated by multiple regression models would prove to be unintuitive, due to the variables used.

In general, multiple regression models in which dependent and independent variables are represented by quantitative data have their results interpreted based on reading the effect that an increase or decrease of one unit in the independent variable causes a change in the dependent variable. This type of interpretation appears to be unintuitive for the purpose of the present study. As an example, in a linear regression between the time in minutes of MVPA and KDE, the result would be read by analyzing the effect that an increase (or decrease) of one KDE unit would have on the time in minutes of MVPA. However, this type of interpretation could cause greater confusion for the reader, suggesting other questions such as “How would you increase a KDE unit or Walkability?”.

This situation could be overcome with regression made from a model with the dependent variable represented by quantitative data and the independent variable represented by dummy variables. This regression was performed and the results were similar to the conclusions observed in this work with the Logistic Regression Analysis. The results of the Multiple Linear Regression showed that adolescents living in KDE Quartiles 3 and 4, determined for radius of 1200 and 1600 meters and with higher walkability values (Quartile 4) had longer MVPA time.

However, the residuals or error terms (observed value – predicted value) of the Multiple Linear Regressions were evaluated according to the assumptions of normality, homoscedasticity and independence. Furthermore, multicollinearity was checked using the vif test (Variance Inflation Factor) between the variables included in the model (FÁVERO and BELFIORE, 2017). Since all the assumptions of the linear regression model estimated by OLS were not met, it was decided to use the Logistic Regression model.

Source: Wooldridge, J.M. (2009) Introductory Econometrics: A Modern Approach. 4th Edition, South-Western College Publishing, Cengage Learning, Boston.

10. Also conduct an analysis of data loss and discuss its potential impact on the associations presented in your study.

R. We recognize that we made a mistake when reporting in the results that 58 adolescents were excluded from the study. In fact, 17 students were excluded from the study. The correct information has been inserted into the text (Page 16, Lines 385 -386).

“Data were collected from 309 adolescents, from these 17 were removed because they lived in census sectors of rural areas.”

It is important to clarify that all statistical analyzes were carried out with a sample size of 292 students. We collected information from 309 students, a number greater than the minimum sample (N = 305) initially established by the sample calculation. However, when we carried out the initial analyses, 17 students did not have their information analyzed because they lived in census tracts in rural areas and, therefore, were removed from the study. Thus, the study sample in which all analyzes were carried out was 292 students.

Recognize that we lost 17 adolescents (5.5%) from the sample because they lived in census sectors of rural areas. We sought advice from a statistician in relation to the missing data. He advised us that while not ideal, 5.5% was not exceedingly high and does not necessitate imputation. We have also included a statement acknowledging the exclusion of students as a limitation in the discussion (Page 27, Lines 584 - 586).

“Finally, the exclusion of the students that live in census sectors of the rural areas, because of the lack of data of these regions.”

11. Finally, to aid reader comprehension of your results, please provide one or more detailed maps of the study area. Additionally, insert explanatory images, such as those from Google tools. 

R. A map was created with more information regarding the study area and is shown as the Supplementary Figure 1. The map was mentioned in the text in the "Methods" section (Characterization of the Study Area) when the study area is described (Page 7, Lines 173).

 “In Supplementary Figure 1 it is possible to visualize the study area.”

Reviewers' comments:

Reviewer #1: 

Introduction:

The study at hand is notable for its comprehensive compilation of evidence. However, it is suggested that the authors consider a more direct and efficient approach, as the paper's current readability leaves room for improvement. The reader's experience can be enhanced by providing a clear direction, such as addressing questions like:

• Why is research examining the distribution density of places for physical activity (PA) significant in daily academic and professional practices?

R. We seek to add information to the text about the Kernel Density Estimation analysis process in order to make clear its importance for academic and professional practices. It was inserted in the text in Pages 4 and 5, Lines 96-107).

 “The Kernel density estimative involves placing a symmetric surface over each point, evaluating the distance from the point to a reference place based on a mathematical function, and summing the value of all surfaces for that reference location. This procedure is repeated for all reference places [27]. The density estimative creates a statistical surface so that, for example, there is an accessibility value measured by the density of the destination, mapped at each point in the study area. It is typically considered a more refined spatial statistical model as it can provide an accessibility estimate for each point in the study area, and not just a binary response of “presence” or “absence”. Therefore, kernel density estimative constitutes a tool that can assist researchers in analyzing the density (concentration) of AF places in a given region and thus verify the possible influence on the PA of individuals. This information can be important for urban planning in order to provide more structures for the practice and encouragement of PA.”

• What is the concept of "walkability" in straightforward terms?

R. We agree that when describing the walkability index in the text, its definition was not very clear. In this way, a simple walkability concept was inserted into the text (Page 3, Lines 69 - 71).

“In this context, walkability is an important characteristic of the urban environment and measures how inviting an area is for pedestrians to access different parts of the city on foot or by bicycle [12, 13].” 

• How do these concepts lead to hypotheses regarding the complex associations between the distribution density of places for PA, walkability, and PA levels?

R. When reading the reviewer's comment, we believe that perhaps the writing of the objective of the study was not very clear, which may have led to confusion in understanding the way in which the analyzes were conducted. Two associations were tested, the first between KDE and PA level and the second association between walkability and PA level. Thus, we seek to make the objective clearer.

“Abstract” Section (Page 2, Lines 35 – 39)

“This study had two objectives: 1- using kernel density estimative, it investigated whether individuals living near PA places that are more intensely distributed than dispersed are more likely to be sufficiently active; 2 - checked whether adolescents who live in neighborhoods with better walkability have a greater chance of being sufficiently active.”

“Introduction” Section (Page 6, Lines 134 – 138).

“This study had two main objectives: 1- using kernel density estimative, it sought to investigate whether individuals living near PA places that are more intensely distributed than dispersed are more likely to be sufficiently active; 2 - check whether teenagers who live in neighborhoods with better walkability have a greater chance of being sufficiently active.”

Furthermore, we added a paragraph in the introduction in which we set out the study's hypotheses. (Pages 5 and 6, Lines 127 – 133)

“We believe that density estimation analyzes can help answer whether the intensity of destinations (PA places) near participants' homes is related to their PA level. Likewise, the walkability index can help to understand the relationships between adolescents' PA level and walking structures in the places where they live. We hypothesized that increasing levels of destination intensity (PA places), measured by KDE and neighborhoods with greater walkability, would be directly associated with the level of PA”.

Methods:

• A noteworthy point arises in lines 186 to 200 where the concept discussed could be better placed in the introduction. This concept is not merely a variable but a way to perceive streets, squares, parks, avenues, and more.

R. We agree with the reviewer that the information described about the walkability index in the methods section is important and should be better presented in the introduction. Thus, the ideas were adjusted and moved to the introduction (Page 3, Lines 71 -79).

“Different walkability indexes were studied to verify the influence of the built environment in the active behaviors of adolescents [14 – 17]. 

However, indexes developed specifically for adolescents were not found, most of the studies adapted the index developed by Frank et al. [12] in the studies developed with the pediatric population. In Brazil, there is no evidence of the existence of an index that can be applied in the whole national territory with unrestricted access data and low operational difficulty [18]. This variety of indexes has been justified by the fact that both the attributes and their quantification for creating the walkability index must be thought specifically for the target population [19].”

Results (general comments):

• Regarding my knowledge of physical activity classifications and concepts, it is crucial to avoid categorizing adolescents as "inactive." None are entirely devoid of physical activity; rather, their level of physical movement relates to health indicators. This implies that the classification should indicate whether the total physical activity was sufficient or insufficient for health. There is evidence suggesting a need for greater precision in these conceptual definitions.

R. We rethought the term used in the manuscript and chose to modify the terms “active” and “inactive” to “Sufficient PA” and “Insufficient PA”. Changes were made throughout the manuscript and are highlighted in yellow.

• The tables provided appear rather convoluted. Given my prior experience in writing articles on environmental factors and physical activity, it is advisable, in my opinion, to enhance the article's clarity to attract readers of all backgrounds. To achieve this, it is recommended to present the tables with: A description of the variables.

R. To increase the clarity of the article, tables 1 and 2 were redone and present a better description of the variables. The new tables can be found in Table 1 (Pages 19 and 20) and Table 2 (Pages 22 and 23).

• Figures illustrating the assessment method and results.

R. We did not find it necessary to insert another figure in the study to illustrate the evaluation method and results. Seeking to remedy this flaw, we rewrote the section explaining the results in Figure 2, providing more details about the estimation of densities and how this is reflected. (Pages 16 and 17, Lines 389 - 409)

“The Figure 2 presents the maps with the Kernel density estimative values of the PA places divided into quartiles for radius of 400, 800, 1200 e 1600m. To produce the image, a kernel with cone radius of 400, 800, 1200, and 1600m was placed over each PA place in the data set. Overlapping cones were added to produce a continuous surface, with closer destinations (PA place) producing higher kernel density estimates. It is important to note that kernel density estimates were calculated independently of adolescents' homes. Kernel density values were extracted so that each participant's home location was assigned the kernel density value of the output cell in which they resided. Although estimates are calculated based on the proximity of destinations (PA place) to each other, the values extracted at each home location provide an indication of the proximity and density of destinations in relation to the home location. Thus, the map illustrates that high kernel density estimates indicate high concentration (intensity) of destinations (PA place) indicating greater proximity between the respondent's home and destinations, as observed in Figures 2C and 2D (the map of kernel covers a large part of the study area and with the areas of each quartile delimited, continuously and without irregularities). Low kernel density estimates indicate insignificant and dispersed PA places, as observed in Figure 2A (absence of continuity in the red areas). Finally, moderate kernel density estimates indicate dispersed PA places or occur when an adolescent's residence is located some distance from a set of highly clustered PA place (Figure 2B). The Supplementary Figure 2 presents the distribution of adolescents in the Kernel Maps, with the density estimative values divided into quartiles for radius of 400, 800, 1200 e 1600m.” 

• The proposed associations as outlined in the study's objectives.

R. By redoing tables 1 and 2, we tried to make the analyzes clearer as the objectives were outlined.

• There seems to be no justification for an excessive number of tables in a study focusing on the association between just three variables, even if they are latent variables. 

R. In fact, the study presents five tables, two of which are complementary. We chose to leave the two tables that present the main results of the study in the body of the manuscript. The table that was previously “Table 2” was transformed into Supplementary table 2, as it presents information on the estimated density values, which were inserted as a curiosity. Thus, the study ended up with two main tables and 3 supplementary tables.

The complementary tables were inserted in the study to resolve possible curiosities of readers regarding the descriptive measures of the variables related to the walkability index (Supplementary Table 1) and some descriptive characteristics of the sample (Supplementary Table 3).

• Lines 414-426 contain arguably the most significant study results. While the text provides a wealth of information, it is essential to address how these findings translate to the real world. Do increased distribution densities of places for PA enhance walkability? Does enhanced walkability correlate with increased physical activity among Brazilian adolescents? Is walkability a mediator or moderator in the positive relationship between the distribution density of places for PA and physical activity levels? The description of results in the tables should aim to clarify these questions.

R. As mentioned above, table 2 was redone in order to clarify the issues raised in the objectives (Pages 22 and 23). Furthermore, when reporting the results of the main analyzes we seek to provide greater detail.

These descriptions are highlighted in the text, in the “Results” section (Page 21, Lines 466 - 481).

Table 2 presents the associations between the PA level of adolescents with Kernel density estimates (radius of 400, 800, 1200 and 1600m) and with the walkability index. The model adjusted for the SS confounding factor showed that this variable was not significant, indicating that the relation between the variables of interest was not significantly affected by SS. Therefore, were presented the results of the analysis of model 1, which was not adjusted for the confounding factor.

Were observed associations between the level of PA and the Kernel density estimative only for the highest intensities at the largest radius places. The increase in the Kernel density estimate for the intensity of the PA places was associated with a probability of being "Sufficient PA" at the Kernel sizes for the radius of 1200 and 1600. The evidence was stronger for quartiles 3 and 4 compared to quartile 1 at 1200m (Q3, OR 2.18 95% CI 1.12–4.22; Q4, OR 2.77 95% CI 1.41–5.43) and 1600m (Q3, OR 3.68 95% CI 1.86–7.30; Q4, OR 3.69 95%CI 1.86–7.30). Furthermore, living in neighborhoods with better walkability was associated with a greater chance of the adolescent being "Sufficient PA". The evidence was stronger for quartile 4 when compared to quartile 1 (Q4, OR 2.58 95% CI 1.33–5.02). 

Discussion:

• To illustrate, I will cite one example, but this issue persists throughout the entire discussion: "The lack of association for the radius of 400 and 800 meters can be explained by the fact that the density produced for these smaller radii can produce estimations with a lot of peaks and discontinuous surfaces depending on the quantity of points observed [50], different from what occurs in the larger radius (lines 513-517)." The discussion section, in addition to being excessively lengthy, is riddled with justifications for unexpected results. The authors frequently attempt to align their study's results with studies that had different objectives, resulting in a confusing and tedious read. It is challenging to discern whether there is a positive, negative, or no association between the analyzed variables up to line 517. The authors appear fixated on discussing descriptive results, which are abundantly available in numerous scientific articles. While it is well-known that many adolescents have insufficient levels of physical activity, this study delves into a complex association, primarily explained numerically, as suggested in the quoted passage. In my academic opinion, the study's strength could lie in its discussion, bridging these intriguing variables with real-world applications, offering valuable insights to readers, especially myself. Unfortunately, the discussion, as currently presented, falls short of these expectations.

R. In order to highlight the importance of our findings and their implications in the real world, we changed the way we discuss our results. All changes are highlighted in the "Discussion" section, starting on Page 24, line 495.

Conclusion:

• Lines 578-583: In my understanding, the conclusion should translate the study's results into a practical and straightforward perspective. It serves as the starting point for a genuine discussion of the study's impact on people's lifestyles.

R. We sought to write the conclusion in a more direct way, highlighting the main findings and how they can impact people's lives, not just the teenagers studied. (Page 28, lines 603 - 610).

“The observed association between the distribution of PA locations and PA at 1200 and 1600m is consistent with our hypothesis that more intensely distributed destinations would be associated with a greater chance of being "Sufficient PA”. As well as living in places with better walkability structure can increase the likelihood of adolescents being "Sufficient PA". 

The results suggest that increasing the intensity of destinations in areas where they are more dispersed; and planning neighborhoods with greater walkability can increase the likelihood of adolescents being “Sufficient PA”.”

---

## [Editor Report · Decision Letter 1]

14 Feb 2024

Built environment and Physical Activity in adolescents: use of the Kernel Density Estimation and the Walkability Index

PONE-D-23-15455R1

Dear Dr. Caetano,

We’re pleased to inform you that your manuscript has been judged scientifically suitable for publication and will be formally accepted for publication once it meets all outstanding technical requirements.

Kind regards,

Vinícius Silva Belo

Academic Editor

PLOS ONE

Additional Editor Comments (optional):

Congratulations!
---

## [Editor Report · Acceptance letter]

8 Mar 2024

PONE-D-23-15455R1 

PLOS ONE

Dear Dr. Caetano, 

I'm pleased to inform you that your manuscript has been deemed suitable for publication in PLOS ONE. Congratulations! Your manuscript is now being handed over to our production team.

Kind regards, 

on behalf of

Dr. Vinícius Silva Belo 

Academic Editor

PLOS ONE